# Probiotic Potential of a Folate-Producing Strain *Latilactobacillus sakei* LZ217 and Its Modulation Effects on Human Gut Microbiota

**DOI:** 10.3390/foods11020234

**Published:** 2022-01-16

**Authors:** Manman Liu, Qingqing Chen, Yalian Sun, Lingzhou Zeng, Hongchen Wu, Qing Gu, Ping Li

**Affiliations:** 1College of Food Science and Biotechnology, Zhejiang Gongshang University, Hangzhou 310018, China; Liumm22@outlook.com (M.L.); chenqingqing@taiyi-nb.com (Q.C.); 15268739569@139.com (Y.S.); q593083982@163.com (L.Z.); wu274086194@outlook.com (H.W.); 2Key Laboratory for Food Microbial Technology of Zhejiang Province, Hangzhou 310018, China

**Keywords:** folate, gut microbiota, *Latilactobacillus sakei* LZ217, probiotic potentials, tolerance

## Abstract

Folate is a B-vitamin required for DNA synthesis, methylation, and cellular division, whose deficiencies are associated with various disorders and diseases. Currently, most folic acid used for fortification is synthesized chemically, causing undesirable side effects. However, using folate-producing probiotics is a viable option, which fortify folate in situ and regulate intestinal microbiota. In this study, the folate production potential of newly isolated strains from raw milk was analyzed by microbiological assay. *Latilactobacillus sakei* LZ217 showed the highest folate production in Folic Acid Assay Broth, 239.70 ± 0.03 ng/μL. The folate produced by LZ217 was identified as 5-methyltetrahydrofolate. LZ217 was tolerant to environmental stresses (temperature, pH, NaCl, and ethanol), and was resistant to gastrointestinal juices. Additionally, the in vitro effects of LZ217 on human gut microbiota were investigated by fecal slurry cultures. 16S rDNA gene sequencing indicated that fermented samples containing LZ217 significantly increased the abundance of phylum Firmicutes and genus *Lactobacillus*, *Faecalibacterium*, *Ruminococcus* 2, *Butyricicoccus* compared to not containing. Short-chain fatty acids (SCFAs) analysis revealed that LZ217 also increased the production of butyric acid by fermentation. Together, *L. sakei* LZ217 could be considered as a probiotic candidate to fortify folate and regulate intestinal microecology.

## 1. Introduction

Folic acid (namely vitamin B9) is one of the water-soluble B vitamins, which is an essential nutrient for humans. The folate is a cofactor in several major metabolic pathways, including the conversion of amino acid and nucleotide synthesis in healthy and proliferating cells [1,2,3]. Unsurprisingly, abnormal folate metabolism is causally related to numerous diseases [4]. A variety of research studies have documented the effects of folate deficiency on birth defects, it is folate deficiency that has been considered a crucial factor of neural tube defects [5]. Besides, folate deficiency has also been linked with increased cancer risk. Low folate homeostasis may lead to reduced DNA methylation, which promotes cancer of proliferating cells in the colon mucosa, supporting rapid and continuous epithelial renewal. The recommended daily intake for adults approved by the European Union is 400 μg/d [6]. Many plants, fungi, and bacteria are capable of synthesizing folate; nonetheless, humans and other mammals cannot synthesize folic acid de novo [7], these cofactors must be derived from exogenous dietary sources, such as vegetables, milk, and fermented dairy products. Unfortunately, the majority of folic acid used for fortification is synthesized chemically, which can cause undesirable side effects such as masking vitamin B12 deficiency [8]. The main derivatives of microbially synthesized folate are tetrahydrofolate, 5-methyltetrahydrofolate (5-MTHF), and 5-formyltetrahydrofolate. Among them, active folate, namely 5-MTHF, is stable folate, which can enter the circulation directly without metabolism and be absorbed and utilized by the human body. Regardless of whether the body takes natural folic acid or synthetic folic acid, it will eventually be converted to 5-MTHF, which is the main form of folate in cord blood. Patients with folate metabolic disorder can either obtain folate through it being absorbed or consumed, and there is no upper limit of active folate, which can be absorbed and utilized by the body, and it will not mask the deficiency of vitamin B12 [9]. In contrast, in situ fortification using folate-producing microbes is a viable option.

Utilizing food-grade bacterial fermentation is a preferred strategy for increasing the folate content and nutritional value of foodstuffs. Folic Acid Assay Broth is usually used to screen folate-producing bacteria. The total folate content of the bacterial fermentation broths and foods can be determined by microbiological assay [10]. The HPLC-MS method is able to discriminate folate derivatives [11]. Many lactic acid bacteria (LAB) strains have been demonstrated to be capable of producing folate, including *Lactococcus lactis*, *Streptococcus thermophilus*, and *Lactiplantibacillus plantarum*. However, the folate production capacity of different strains varies greatly. For example, no other *Lactobacillus* species has been shown to be as capable to produce folate except for *Lactiplantibacillus plantarum* [12]. LAB strains that produce large numbers of folate and survive in the gastrointestinal tract can be regarded as effective probiotics to resist folate deficiency. Currently, several LAB strains or folate-producing *Lactobacillus* are preliminarily applied in fermented food products to improve the folate concentrations [13]. Among fermented products, fermented milk is supposed to be a potential substrate for folate fortification, since the folate-binding protein in milk improves the stability of folate, the bioavailability of 5-MTHF, and other tetrahydrofolates may also be increased [14]. The research by Crittenden et al. suggested that the fermentation of skim milk with *Streptococcus thermophilus* alone can elevate folate levels, and when fermented in combination with *Bifidobacterium animalis*, significantly increased folate concentrations to more than six times [15]. Although the folate state in fermented foods is due to the presence of LAB strains, it is still relatively low in relation to the recommended daily intake for an adult (400 μg/d). Thus, it is of great significance to screen bacteria with high folate-producing ability, which could be used as folate fortifiers to replace chemically synthesized folic acid.

In recent years, strategies utilizing probiotics to improve and regulate the intestinal microbiota have received substantial attention. When administered in adequate amounts, probiotics can modulate the intestinal ecosystem, improve immunity, and confer health benefits on the host [16]. Previous research has proposed that folate-producing LABs colonized in the gut become a stable folate supplier to improve folate deficiency of the host [17,18]. However, it is rarely reported that folate-producing bacteria can modulate gut dysbacteriosis. The latest study by Zhang et al. confirmed that administration of *Lactobacillus plantrum* GSLP-7 V and its fermented yogurt restored the key genera (such as *Prevotella*, *Dialister*, *Peptostreptococcus*, *Streptococcus*) and recovered the serum folate to normal levels in folate-deficient rats [17]. Therefore, studying the interaction between folate-producing *Lactobacillus* and hosts’ microbiota can more comprehensively clarify its modulation of gut dysbacteriosis. Ideally, the experiment in humans is the most effective and direct method; however, collecting gut microbes of human volunteers is not always ethically and economically feasible. Generally, the results of in vitro experiments can be partially representative of the findings in human volunteers [19]. Moreover, in vitro models that mimic human colon ecology offer the possibility to study the interaction of probiotic strains with the intrinsic colonic microbiota, such as microbial composition and metabolism (production of volatile fatty acids, tryptophan derivatives).

In the present study, we isolated six folate-producing *Lactobacillus* strains from raw cow’s milk. We further investigated the probiotic properties of an isolate (LZ217) with the highest folate production, including tolerance to environmental stresses (temperature, pH, NaCl, and ethanol) and resistance to gastrointestinal juices. To explore the effects of LZ217 on human gut microbiota, the interaction between LZ217 and the intestinal microbiota of fresh human feces was investigated using an in vitro simulated intestinal fermentation model. The effects of LZ217 on gut microbiota structure and the content of short-chain fatty acids (SCFAs) were also studied.

## 2. Materials and Methods

### 2.1. Isolation of Folate-Producing Lactobacillus Strains

A total of 21 *Lactobacillus* strains were isolated from 40 raw milk samples, provided by the ranch New Hope in Hangzhou, China. Each 25 mL milk sample was suspended in 225 mL of sterilized 0.9% saline and a series of dilutions were prepared and applied to the surface of MRS agar plates (mixed with 2% CaCO_3_), incubating for 48 h. The bacterial colonies were isolated at random, which formed clear zones around it by dissolving CaCO_3_. Then each isolated strain was incubated overnight in MRS broth for without shaking. Different culture cells were collected, washed, and resuspended with sterilized 0.9% saline. Subsequently, the obtained suspensions were inoculated into Folic Acid Assay Broth, incubated overnight, and transferred several times in the Folic Acid Assay Broth. During incubation, each tube was wrapped in aluminum foil. Those with OD600 > 0.5 were initially identified as folate producers after eight rounds of growth. All bacteria mentioned were cultured at 37 °C. All culture mediums used were purchased from Luqiao, China.

The folate-producing ability of 6 isolates was evaluated by microbiological assay [20] to screen for LAB with efficient folate-producing ability. Cryoprotected *Lacticaseibacillus rhamnosus* ATCC 7469 (purchased from Guangdong microbial species preservation Center), an auxotrophic strain for folate, was used in the microbiological test to quantify the folate and its derivatives. Each strain was fermented overnight in Folic Acid Assay Broth. The supernatant and cell extracts were respectively obtained according to the method described by Koontz et al.: by centrifugation, then measuring the extracellular and intracellular folate content [21]. Various concentrations of folate standard ranging from 0.2 ng/mL to 0.8 ng/mL were prepared for standardization. The reaction mixture included a working buffer (0.01% sodium ascorbate in 0.1 M sodium phosphate buffer pH 6.8), different concentrations of folate standard solutions, double concentrated Folic Acid Assay Broth, and sterile water. The Folic Acid Assay Broth and diluted samples were mixed into a 96-well plate, which was internally coated with *Lacticaseibacillus rhamnosus* ATCC 7469. After the addition of the folic acid standard, the bacteria grew until the folic acid was completely consumed. These reagents were cultured in the dark for 48 h, and the growth was determined at 630 nm using a SpectraMax190 microplate reader (Microplate Reader, Molecular Devices, San Jose, CA, USA).

The isolate LZ217 with efficient folate-producing ability was further identified by physiological and biochemical assays, followed by 16S ribosomal DNA (rDNA) gene sequencing, and constructing the phylogenetic tree within the MEGA 5.1 package.

### 2.2. Analysis of 5-Methyltetrahydrofolate by LC-MS

To further determine whether 5-MTHF is present in the folate derivatives produced by *L. sakei* LZ217, the chromatographic analysis was performed using an Agilent Technologies LC-MS system (Agilent Technologies, Santa Clara, CA, USA). A folate standard, 5-MTHF, was procured from Sigma and used after dissolving. The isolated strain was incubated overnight in Folic Acid Assay Broth; fermentation broths were extracted by centrifuging treatment for LC-MS analysis. The inoculated medium before culture was used as a blank control. Solvents were 0.1% (*v*/*v*) HCOOH (solvent A) and acetonitrile (solvent B). The samples were subjected to gradient elution using the following procedure (% (*v*/*v*) mobile phase B): from 0–3 min, 10%; 3–5 min, 10–15%; 5–8 min, 15%; 8–13min, 15–50%; and 13–16 min, 50–10%, followed by isocratic elution at 10% B for 9 min.

### 2.3. Environmental Stress Tolerance Assay

The ideal strain with industrial potential should resist adverse conditions [22]. Thus, it would be necessary to evaluate the tolerance of LZ217 to extreme temperatures, presence of weak acids, osmotic stresses, and ethanol. In brief, to study the effect of several stress factors on LZ217, the strain was inoculated at 1% (*v*/*v*) in 10 mL of the MRS broth and incubated at 37 °C overnight. Fresh cells during stationary phases were collected by centrifugation (10,000 g, 8 min), washed at least three times, and resuspended in PBS (pH 7.2), for later tolerance assay use. The number of viable fresh cells was determined and diluted to about 2.5 × 10^9^ CFU/mL before the assays. All the experiments were repeated three times.

For temperature stress tolerance assay, the bacterial suspension prepared as above was inoculated at 1% (*v*/*v*) into 10 mL of MRS broth and incubated in different temperatures (4 °C, 15 °C, 25 °C, 37 °C, and 42 °C). The growth of LZ217 was evaluated by measuring OD600 overnight. All the experiments were repeated three times.

To test the stress response features, a variety of MRS broths were prepared with different pHs (2.0, 3.0, 4.0, 5.0, and 6.0) and varying concentrations of NaCl (3%, 6%, and 8%) or ethanol (2.5%, 5%, and 10%) [23]. The bacterial suspension was prepared as above with 100 μL transferred into 10 mL of various MRS broths. Following incubation, 1 mL was removed from each fermented broth and diluted appropriately. Then, the total viable counts of cells were measured by the plate count method. All the experiments were repeated three times.

### 2.4. Simulated Gastrointestinal Tolerance Assay

The tolerance of *L. sakei* LZ217 to simulated gastrointestinal juices was studied as described by P. Li et al. [23]. In order to prepare artificial gastric juice, pepsin (Sigma–Aldrich, St. Louis, MO, USA) was dissolved in sterile PBS to a final concentration of 0.3% (*w*/*v*), and the mixture was adjusted to pH 2.0, 3.0, and 4.0. Similarly, simulated intestinal juices were prepared by suspending 1 g trypsin (Sigma–Aldrich) in sterile PBS, containing 0.1%, 0.3%, and 0.5% pig bile salts (Sigma–Aldrich), respectively. A 400 μL bacterial suspension, as mentioned in Section 2.3, was mixed with 600 μL NaCl (0.5% *w*/*v*) and 2 mL simulated gastric juice and simulated small intestinal juice, respectively. A fermented liquid containing 200 μL was removed from each sample after 37 °C of incubation at 0 h, 2 h, and 4 h, then diluted appropriately and determined for the viable counts of cells on MRS plates. All the experiments were repeated three times.

### 2.5. The In Vitro Effects of Lactobacillus on Human Gut Microbiota in the Fecal Slurry Cultures

#### 2.5.1. Study Design and Sampling

Fecal samples were collected from 10 healthy human volunteers (the following paragraphs are denoted by 1, 2, 3, 4, 5, 6, 7, 8, 9, 10) living in Hangzhou, China, ranging in age from 22 to 35 years old. These volunteers were in good health and had not been taking any medications and antibiotics in the previous 3 months. All donors provided their informed, written consent, and the study was supported by the Ethics Committee of the Zhejiang Gongshang University and Zhejiang Academy of Agricultural Sciences (Zhejiang, China). The fresh fecal samples were collected immediately from these volunteers after defecation. Each 0.2 g sample was homogenized in 0.1 M phosphate-buffer solution (PBS, pH 7.2) to generate 10% (*v*/*v*) fecal suspension. After filtration, the different supernatants were transferred to a new tube and stored anaerobically; the other section of each sample was stored at −80 °C for later use.

The fermentation model utilized was based on the protocol of Rycroft et al. [24] and Lei et al. [25] with some modifications. The VIS medium contained the following: starch (8 g/L), tryptone (3.0 g/L), peptone (3.0 g/L), NaCl (4.5 g/L), bile salts No. 3 (0.4 g/L), L-cysteine hydrochloride (0.8 g/L), hemin (0.05 g/L), KCl (2.5 g/L), MgCl_2_·6H_2_O (0.45 g/L), CaCl_2_·6H_2_O (0.2 g/L), yeast extract (4.5 g/L), KH_2_PO_4_ (0.4 g/L), 1 mL Tween 80, and 2 mL of a solution of trace elements. The above reagents were purchased from Changqing Chemical Co., Ltd. (Hangzhou, China). To assess the effect of *L. sakei* LZ217 on human fecal microorganisms, *L. sakei* LZ217 cells were added to the VIS medium to reach a population of approximately 1 × 10^8^ CFU/mL, which is a VIL medium. The 500 μL fecal supernatant was added to the VIS and VIL medium (5 mL) and fermented anaerobically overnight. The 2 mL samples of fermentation broth were stored at −80 °C, which were used for DNA sequencing and gut microbiota structure. The fermentation broth was centrifuged in a sterile centrifuge tube at 12,000 rpm for 3 min. The separated supernatant was stored at −20 °C, which was used for the determination of SCFAs content.

#### 2.5.2. DNA Extraction and 16S rDNA Gene Sequencing

Bacterial genomic DNA was obtained from fermented manure samples using the QIAamp DNA Stool Mini Kit (cat 51604) following the manufacturer’s instructions (QIAGEN, Hilden, Germany). 16S rDNA gene sequencing for the V3-V4 region of all extracted DNA was conducted by Genetalks Biotechnology Co., Ltd. (Beijing, China). The amplification primers were 338F with 806R, the fragment was 450 bp, and the PCR reaction was repeated at least 3 times for each sample. The PCR products were verified, quantified, and then mixed according to the sequencing requirements of each sample. In Miseq library construction and sequencing, the collected fluorescent signal results were counted to obtain the sequence of the template DNA fragment. The obtained sequences were spliced according to overlap relationship, and the operational taxonomic units (OTUs) clustering analysis was performed after distinguishing the samples.

#### 2.5.3. Detection of Short-Chain Fatty Acids by Gas Chromatography

The contents of SCFAs from filtered frozen supernatants were determined by gas chromatography (GC) as a reported approach with some modifications [26,27]. Briefly, peak identity and quantification were determined using a crotonic acid as an internal standard, including standards of acetic, propionic, and butyric. 500 μL of the filtered supernatant was mixed with 100 μL of crotonic acid, 20 μL mixture was injected into the injection bottle for detection. An InertCap FFAP gas chromatography column (0.25 mm × 30 m × 0.25 μm) was used, and compounds were quantified from the peak areas using the external standard method. The determination program for SCFAs was as follows: the initial column temperature was 70 °C, then increased to 180 °C at the rate of 15 °C/min, and maintained at this temperature for 2 min, thereafter, increasing at a rate of 40 °C/min until reaching 240 °C, which was held there for 3 min. The injection volume of the sample was 1 μL, the temperature of the detector was 250 °C, and the column flow was 2.93 mL/min. Each sample was analyzed three times.

#### 2.5.4. Statistical Analysis

Diversity statistical analyses of gut microbiota were performed using the Visual Genomics-AS Software (Shanghai Infinity Biotechnology Co., Ltd., Shanghai, China). The values of *p* < 0.05 were regarded as statistically significant. All data were shown as mean ± standard deviation (SD). Significant differences among the sample were analyzed using one-way ANOVA, followed by Tukey’s test at *p* < 0.05.

## 3. Results and Discussion

### 3.1. Isolation of Folate-Producing Lactobacillus Strains

Six of the 21 isolates from milk were regarded as folate producers, which expressed growth in the folate-free medium after several rounds of subculture. The folate production of the six isolates was measured by microbiological assay [20]. As shown in Appendix A, folate yield by the six isolates varied from 5.50 ± 0.02 ng/mL to 239.70 ± 0.03 ng/mL. LZ217 exhibited the highest folate production of 239.70 ± 0.03 ng⁄mL. This isolate was chosen for further studies, including the study of folate forms and characterization of the probiotic properties. LZ217 was identified by the physiological and biochemical assays (data not shown) and 16S rDNA sequencing. The sequence analysis showed 98% similarity to *Latilactobacillus sakei*, phylogenetic analysis indicated that LZ217 was closely related to the strain *L. sakei* NBRC 107130 (Appendix A). Therefore, LZ217 was identified as *L. sakei* and named *Latilactobacillus sakei* LZ217.

### 3.2. Analysis of the Folate Forms by LC-MS

As above, LZ217 exhibited the highest folate production. To further confirm whether 5-MTHF is present in the folate derivatives produced by *L. sakei* LZ217, LC-MS analyses were performed. The chromatograms (Figure 1) displayed the presence of 5-MTHF as a form of folate biosynthesized by LZ217. Figure 1A shows the retention time at 5.65 min and Figure 1B shows the molecular masses at 458.1809 Da of the standards 5-MTHF. Figure 1C shows the retention time at 5.86 min and Figure 1D shows the molecular masses at 458.1835 Da of the samples. Compared with the folate standard, we demonstrated that *L. sakei* LZ217 can produce 5-MTHF. 5-MTHF, as natural folate, accounts for 98% of all folates in plasma and does not require conversion by dihydrofolate reductase to be active [8]. Consequently, *L. sakei* LZ217 contained the naturally occurring folate, which has great potential in folate fortification.

### 3.3. Environmental Stress Tolerance Assay

LAB strains considered as probiotics may encounter various environmental stress factors during the fermentation of milk and dairy products (e.g., extreme temperatures, presence of weak acids, osmotic stresses, and ethanol) [23]. We focused on stress response features of LZ217 to various incubation temperatures, pH, different concentrations of NaCl (3%, 6%, and 8%) and ethanol (2.5%, 5%, and 10%). Figure 2A shows the growth of LZ217 under a range of temperatures (4–42 °C). During the fermentation of LAB strains, different fermentation temperatures would affect the production of flavor substances. Ideal probiotic strains should be able to adapt to different temperature conditions. For example, the temperature of yoghurt fermented by strains is usually 42°C and the storage temperature is 4 °C [28]; LZ217 grew at 4 °C and 42 °C, although not as well at 37 °C, indicating it can tolerate well to the temperature of the dairy industry. Additionally, the pH of the fermented yoghurt generally maintains its shelf life at about pH 4. Figure 2B shows that LZ217 still maintained a relatively high survival rate (57.11%) even in pH 4. When a LAB strain produces lactic acid during fermentation, a base will subsequently be applied to prevent excessive pH reduction, which will render the conversion of the free acid to its salt form, thereby increasing the osmotic pressure of bacteria [29]. Therefore, osmotolerance will be a prerequisite for the commercial application of LAB strains. Ethanol is also one of the most common stress factors; too high concentration of ethanol would affect the metabolism and physiological activity of bacteria. The survival rates of *L. sakei* LZ217 in 3%, 6%, and 8% of NaCl were 84.08%, 46.15%, and 5.26% (Figure 2C), respectively, indicating that this strain exhibited growth in appropriate osmotic concentration of NaCl. Moreover, previous research has shown that the majority of LAB strains were tolerant to 5% ethanol [30]. Although the tolerance decreased with increasing ethanol concentration, the survival rate of *L. sakei* LZ217 was still 41.54% in 10% of ethanol (Figure 2D). Together, *Lactobacillus* isolate was capable of surviving harsh conditions.

### 3.4. Simulated Gastrointestinal Tolerance Assay

The effective effect of probiotic strains depends on their ability to survive in the host’s defense system and colonize the gastrointestinal tract [31,32]. Thus, evaluating their tolerance to the gastrointestinal tract (low pH and the presence of bile salts) is a priority. Here, we assessed the survival ability of LZ217 within 4 h of simulated gastric transportation at various pHs (pH 2.0, 3.0, and 4.0) (Figure 3A). It showed that the tolerant ability increased with the rise of pH, *L. sakei* LZ217 had good tolerance at pH 4.0. The survival rates of *L. sakei* LZ217 at 2 h and 4 h of pH 2.0 were 11.53% and 8.54% respectively. Studies reported have shown that most *Lactobacillus* can survive well at pH 3.0, but rarely at pH 2.0. In comparison, *L. sakei* LZ217 can survive better under a harsh gastric juice environment [33]. Moreover, the presence of bile salts will also affect the viability of intestinal microbiota [34]. Bile salts at a concentration of 0.15–0.3% are generally considered as the appropriate concentration of probiotics. In the presence of 0.3% bile salts, the survival rate of LZ217 for 2 h and 4 h was 31.89% and 20.81%, respectively (Figure 3B). LZ217 exhibited a high tolerance to small intestinal transit and did not completely lose viability even after 4 h incubation in simulated small intestinal juices with 0.5% bile salts. In summary, *L. sakei* LZ217 exhibits acid-fastness and salt-tolerance properties.

### 3.5. Evolution of the Gut Microbiota and the SCFAs Concentrations in the Fecal Slurry Cultures

#### 3.5.1. In Vitro Effect of the *L. sakei* LZ217 on the Structure of Gut Microbiota

The bacterial genomic DNA of the fermentation broth was gained and analyzed to comprehend the effect of LZ217 on gut microbiota. The Shannon curves and rarefaction curves were used to evaluate the sequence quality (Appendix A). The top five phyla (Figure 4A) and top 30 genera (Figure 4B) ranked microflora structure of the VIS group and VIL group were listed visually in the chart. At the phylum level, the five phyla detected in all of the samples were Firmicutes, Actinobacteria, Fusobacteria, Bacteroidetes, and Proteobacteria. Overall, bacterial species were similar between the two groups. The gut microbiota of each volunteer had a common feature after fermentation in both media, with more Firmicutes in the VIL medium than in the VIS medium. This result suggested that the addition of LZ217 to the medium probably increased the number of Firmicutes and promoted the growth of other bacteria (such as *Ruminococcus* 2) in Firmicutes. Consistent with this result, Firmicutes became the dominant flora after *Lactiplantibacillus plantarum* ZJ316 inoculation [35].

The sample cluster heatmap analysis (phylum level) with varying volunteers is exhibited in Figure 4C. Consisting of the aforementioned chart analysis results, it can be seen that Firmicutes in each volunteer in the VIL group was markedly varied from that in the VIS group.

At the genus level, the percentages of *Lactobacillus* in the gut microbiota of each volunteer were significantly increased in the VIL group compared to the VIS group. There are two possible explanations for this. The first is that *L. sakei* LZ217 was inoculated in the VIL group, which increased the abundance of *Lactobacillus*. Additionally, previous reports showed that most *Lactobacillus* species, with the exception of *Lactiplantibacillus plantarum*, cannot produce folate [12]. It was speculated that the presence of *L. sakei* LZ217 can supply the adequate folate for folate-consuming *Lactobacillus* to grow. However, the proportion of increase was inconsistent, with *Lactobacillus* of some samples showing very significant, most visually in volunteer 10. The dominant gut microbiota of humans has host specificity and can be divided into three intestinal types, as described by Arumugam et al.: *Bacteroides* (enterotype 1), *Prevotella* (enterotype 2), and *Ruminococcus* (enterotype 3) [36]. As shown in Figure 4B, the dominant bacteria of volunteers 4, 6, and 10 was *Prevotella*, and the others were *Bacteroides*. Overall, the *Bacteroides* in the gut microbiota of each volunteer generally decreased in the VIL medium. But there were exceptions with *Bacteroides* of volunteers 5 and 7, found to only have a slight change. Compared with volunteers with enterotype 1, the gut microbiota of volunteers with enterotype 2 easily changed, and *Lactobacillus* had changed significantly. These results were parallel with previous enterotype studies, the intestinal microbiota in enterotype *Bacteroides* was not easily altered [37]. Moreover, *Prevotella* in the gut microbiota of volunteers 1 and 3 increased.

LEfSe analyses was carried out in both groups in taxon abundance to estimate whether there were remarkable discrepancies (Figure 5). The strikingly enriched bacterial communities in each group were represented by the corresponding nodes in Figure 5A. LEfSe results with a 6.0 LDA score of the VIS and VIL group were exhibited in Figure 5B. At the phylum level, the Proteobacteria of group VIS and the Firmicutes of group VIL were dramatically enriched. At the class level, the Gammaproteobacteria and Chloroplast of group VIS were observably enriched, the former showing greater differences. The Bacilli in group VIL were markedly different. At the order level, the microflora of the VIL group was notably more abundant than that of the VIS group. In the VIL group, g: Lactobacillales, n: Rhodocyclales, t: Pasteurellales, w: Xanthomonadales, and g: Lactobacillales showed significant differences. Only q: Enterobacteriales in group VIS was significantly enriched. At the family level, f: Lactobacillaceae, d: Enterococcaceae, b: Paenibacillaceae, k: Burkholderiaceae, m: Rhodocyclaceae, s: Pasteurellaceae, and v: Xanthomonadaceae showed significant differences in group VIL, and f: Lactobacillaceae showed the most significant difference. Only p: Enterobacteriaceae in group VIS was significantly enriched. At the genus level, e: *Lactobacillus*, c: *Enterococcus*, a: *Brevibacillus*, j: *Cupriavidus*, l: *Methyloversatilis*, r: *Haemophilus*, u: *Stenotrophomonas*, and e: *Lactobacillus* in group VIL were significantly enriched. In descending order of differences, these were o: *Escherichia_Shigella*, h: *Lachnospiraceae*ND3007 g, and i: *Megasphaera* in group VIS. Overall, Firmicutes of the VIL group showed the most significant difference. We proved the addition of *L. sakei* LZ217 promoted the growth of phylum Firmicutes and genus *Lactobacillus*, *Faecalibacterium*, *Ruminococcus* 2, and *Butyricicoccus* compared with the VIS group. Some studies have shown that 87% of intestinal microbiota lack the genes required for folate synthesis, indicating possible effects of folate-producing LABs on the microbiota if they are introduced into the gut [38]. Similar studies showed that *L. plantarum* GSLP-7 V introduced can promote the abundance of the *Faecalibacterium*, thus, Firmicutes became the dominant phylum in the gut [17]. *Butyricicoccus* is an efficient butyrate-producing clostridial cluster IV genus, first isolated from the caecal content of a chicken by Eeckhaut V et al. [39]. The results indicated that LZ217 has modulation effects on human intestinal microbiota and may have great prospects for the future. But further research evaluating the influences of LZ217 on the gut microbiota metabolism of the host utilizing in vivo models will provide additional insight into the interactions between LZ217, enteric microorganisms, and the host.

#### 3.5.2. Effect of LZ217 on Short Chain Fatty Acid Content in the Fecal Slurry Cultures

The SCFAs including acetic acid, propionic acid, and butyric acid of the fermentation samples by the gut microbiota were determined accurately by GC. The molar concentrations of SCFAs after fermented are shown in Figure 6. Figure 6A–C represents acetic acid, propionic acid, and butyric acid contents in the fermentation liquid, respectively. The acetic acid contents from VIS and VIL groups had no significant difference. The content of propionic acid of the VIS group was observably higher than the VIL group (*p* < 0.01). Notably, butyric acid concentrations in the VIL group were much higher than in the VIS group (*p* < 0.01).

Acetic acid, which is able to provide energy for intestinal microbiota, is the major metabolite of numerous bacteria according to previous trials [40]; propionic acid is the most dominant metabolite of *Bacteriodetes*, which reduces the serum cholesterol level [41] and prevents diet-related obesity [42], thus is significant to the health on the host; butyric acid, which is mainly produced by Firmicutes, is also closely associated with intestinal health [43]. Butyric acid, as an essential energy source of enteric epithelial cells, lays the foundation for the modern symbiosis theory between intestinal epithelial cells and microflora [44]; butyrate can be quickly absorbed, stimulate the growth of the small intestinal epithelium, and promote wound healing, while it dampens proliferation in colonic cancer cells [45]. Reportedly, the production of SCFAs was related to the species and quantities of gut microflora, *Alistipes* can promote the production of acetic acid [46], *Bacteroides* favorably promotes propionic acid production [47], *Roseburia*, *Faecalibacterium*, *Ruminococcus*, and *Butyricicoccus* havehealth-associated genera with a positive potential of producing butyric acid [48,49]. We demonstrated that butyric acid significantly increased, whereas acetic acid and propionic acid tended to be decreased due to the addition of *L. sakei* LZ217 in fecal slurry culture fermentation compared with the VIS group. This may be attributable to the fact that *L. sakei* LZ217 altered the abundance of gut microflora. The increase of phylum Firmicutes and genus *Lactobacillus*, *Faecalibacterium*, *Ruminococcus* 2, and *Butyricicoccus* may promote the production of butyric acid. The decrease of *Bacteroides* and *Alistipes* may directly affect the concentrations of acetic acid and propionic acid. In short, LZ217 can promote the production of butyric acid by the gut microbiota, which plays a critical role in maintaining healthy homeostasis.

## 4. Conclusions

Folate is essential to metabolic function and the health of its hosts. LAB strains, as excellent candidates, can supply folate to human hosts due to their advantage of in situ fortification. *Lactobacillus* has benefits for human health, which is considered in the majority of critical functional foods. Preliminary investigation of a novel probiotic strain is vital to verify their qualification as good probiotics. In this study, a folate-producing strain, *L. sakei* LZ217, as a potential folate in situ producer for the food industry, was isolated from raw cow’s milk. The folate produced by LZ217 was identified as 5-MTHF by LC-MS; its probiotic properties were investigated showing that it can survive in vitro environmental stresses (NaCl and ethanol) and simulated human gastrointestinal tract conditions. On the other hand, based on in vitro fermentation studies, we demonstrated the interaction between LZ217 and human gut microflora, which regulates human gut microflora and promotes the production of butyric acid. The addition of *L. sakei* LZ217 promoted butyric acid production after sample fermentation and altered the abundance of gut microflora in human fecal microbiota. However, whether the effect of this strain on gut microbiota is related to human enterotype or not needs to be further explored with the participation of more volunteers.

In conclusion, *L. sakei* LZ217, as an excellent candidate, has high folate production and superior traits, and may in situ enhance the folate state. It has considerable probiotic potential by promoting intestinal health. In the future, additional studies, including animal models and human populations, could be combined with molecular biology techniques to confirm whether folate produced by *L. sakei* LZ217 exerts health benefits by modulating the gut microbiome.

## Figures and Tables

**Figure 1 foods-11-00234-f001:**
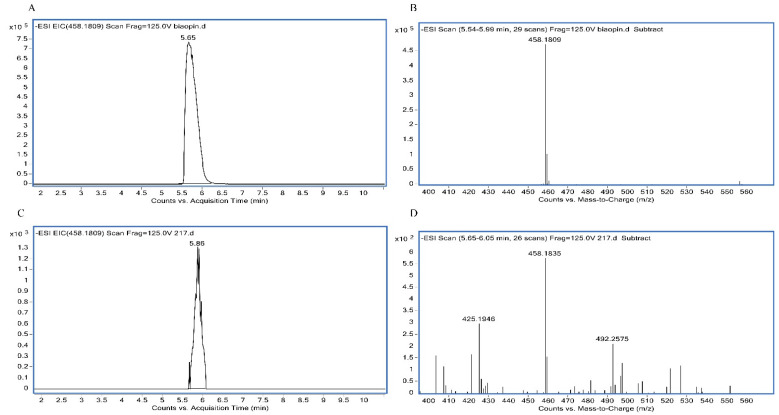
Identification the form of folate produced by *L. sakei* LZ217 using LC-MS chromatogram. (**A**,**B**) respectively showed the retention time and the molecular masses of the standards 5-MTHF. (**C**,**D**) respectively showed the retention time and the molecular masses of the cell extracts from *L. sakei* LZ217.

**Figure 2 foods-11-00234-f002:**
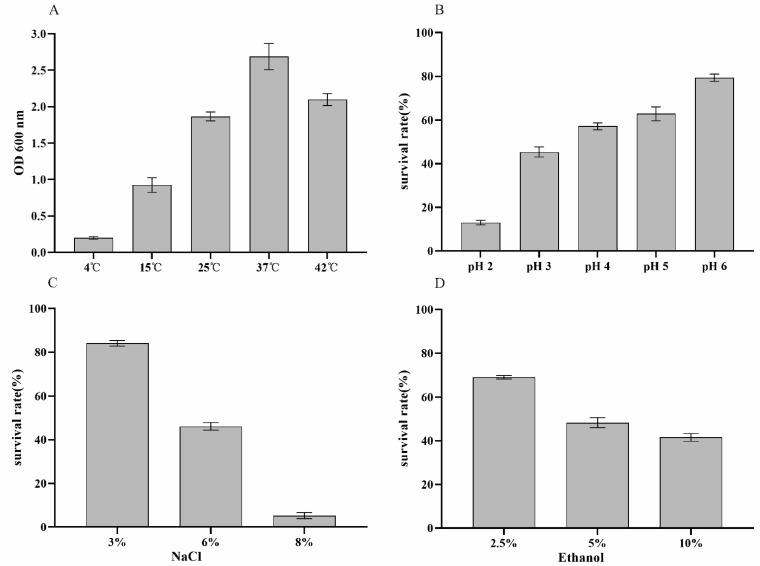
Tolerance of *L. sakei* LZ217 to different environmental stresses. (**A**) Different temperature (4 °C, 15 °C, 25 °C, 37 °C, and 42 °C), (**B**) different pHs (2.0, 3.0, 4.0, 5.0, and 6.0), (**C**) NaCl (3%, 6%, and 8%), and (**D**) ethanol (2.5%, 5%, and 10%). The growth was evaluated by measuring OD600 after 24 h of different incubation temperature. The survival rate was evaluated by plate count after 48 h of incubation at 37 °C.

**Figure 3 foods-11-00234-f003:**
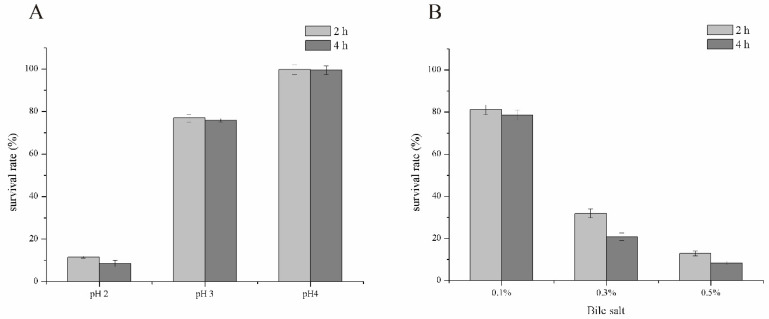
Survival rate (%) of *L. sakei* LZ217 (**A**) in simulated gastric juice at different pHs and (**B**) during simulated small intestinal transit under different bile salt concentrations.

**Figure 4 foods-11-00234-f004:**
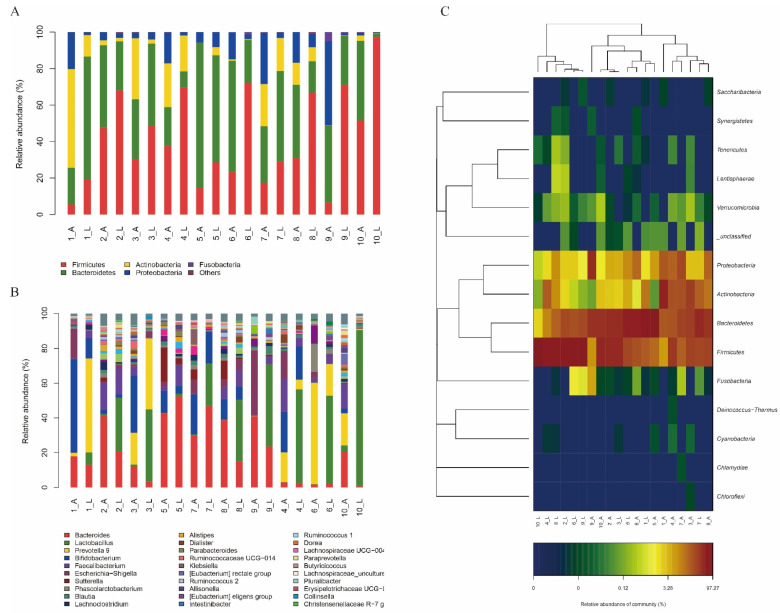
Relative abundance of intestinal microbiota at phylum level (**A**) and genus level (**B**), and the community heatmap (phylum level) (**C**) of intestinal microbiota fermented in different media. Note: 1–10 indicates volunteer number, “_ A” for the VIS group, “_ L” for VIL group.

**Figure 5 foods-11-00234-f005:**
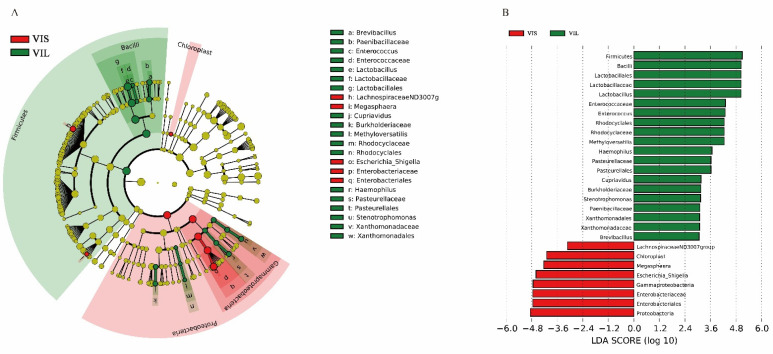
Analysis of different abundant bacterial taxa using LEfSe. (**A**) The evolutionary branching diagram. The circle from inside to outside represents the classification level from phylum to species, and the dot (yellow: no significant difference, other colors: significant difference) represents a classification at this level. (**B**) The histogram of LDA value distribution and the length of the histogram represents the impact of different species.

**Figure 6 foods-11-00234-f006:**
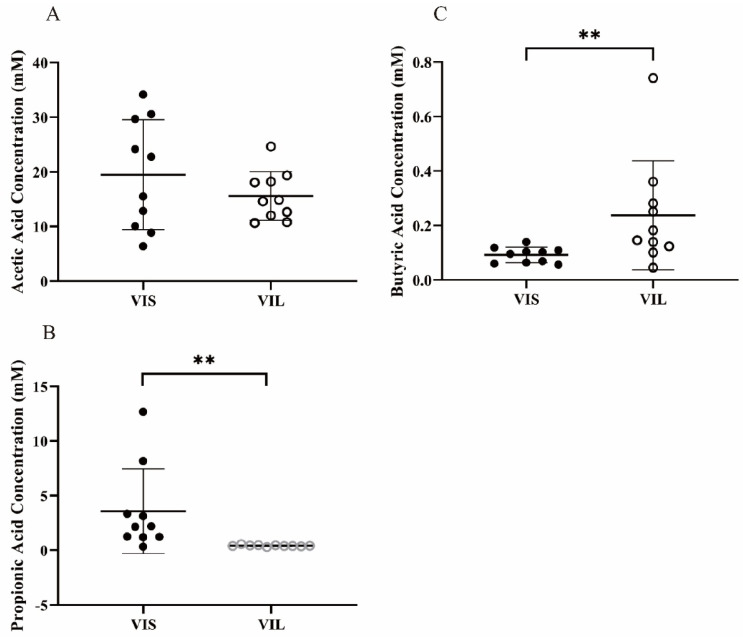
The molar concentrations of SCFAs produced during the fermentation. (**A**) Acetic acid concentration; (**B**) propionic acid concentration; (**C**) butyric acid concentration. The fermentation time was 24 h. The horizontal coordinates are a different medium. ** *p* < 0.05 with two stars.

## Data Availability

The datasets generated for this study are available on request to the corresponding author.

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
