# Peer review of "Probiotic Potential of a Folate-Producing Strain Latilactobacillus sakei LZ217 and Its Modulation Effects on Human Gut Microbiota"

_foods, 2022, doi:10.3390/foods11020234_

Round 1

Reviewer 1 Report

Liu et al. have presented a manuscript entitled “Probiotic potential of a folate-producing strain Lactobacillus sakei LZ217 and its modulation effects on human gut microbiota”. The work focused on the selection of a strain with the ability to produce high amounts of folate. Furthermore, the selected strain LZ217 was tested in order to investigate its potential probiotic properties, including tolerance to salt stress and ethanol, resistance to gastrointestinal juices, effect on the human intestinal microbiota and short-chain fatty acids content.

Although the topic of the work is of interest, I believe that the manuscript has several weaknesses.

Introduction:

The introduction does not provide sufficient background. The topics regarding the selection criteria of probiotic strains and the effect of probiotic on the modulation of human gut microbiota have not been taken into account.

Materials and methods are not clear.

In particular:

Section 2.1

The authors should review this section and better describe the screening activity that led to the selection of the LZ217 strain. For example: how many milk samples were processed? Where did the samples come from? How many isolates were screened for folate production?

It is not clear which medium was used to assess folate production. Folic acid assay broth is reported in section 2.1 and skim milk medium in the abstract.

Section 2.3

There are several stress factors that a strain may encounter during the fermentation of milk and dairy products (e.g. low and high temperature, presence of weak acids etc.). Why did the authors decide to consider tolerance to ethanol stress as a criterion for strain selection and not consider the other stress factors? It would be more appropriate to evaluate the tolerance of the strain to other stress factors rather than ethanol.

Furthermore, the cells used in stress tolerance assay were in what physiological state? The response to a stress condition can vary between cells in exponential and stationary phases.

It is also not clear whether the authors intend to assess growth or survival of the strain under stress conditions. In section 2.3 they refer to growth, whereas in the results (section 3.3) they assess survival.

The concentration of the inoculum has not been reported.

Section 2.4, line 145

What is meant by 'pretreated cells'? That they were first subjected to NaCl or ethanol stress, and then subjected to the grastrointestinal transition? Please explain further.

The concentration of the inoculum has not been reported.

Results and discussion:

Section 3.5.2

The title of the paragraph is not appropriate because the authors did not study the evolution of SCFAs, but only compared the concentrations at the end of fermentation. The title should be replaced.

Minor remark

Please replace the term “flora” by microbiota

Please write the name of the microbial species in italics

The authors should use the new taxonomy of the genus Lactobacillus (https://doi.org/10.1099/ijsem.0.004107)

Reviewer 2 Report

The aims of the study are clearly formulated.

Regarding the experimental design, the authors mention, for example for the analysis of short-chain fatty acids by GS, that the experiments were repeated twice. What about other experiments/analyses, what was the experimental design?

lines 301-302: "At the genus level, the percentages of Lactobacillus in the gut flora of each volunteer 301 were significantly increased in the VIL group compared to the VIS group." Why? Any possible explanations?

line 337: Are these "modulation effects" of LZ217 consistent and coherent?

Did the authors analyse the different fermentation broths after fermentation to check whether they could recover the Lactobacillus sakei strain LZ217? The higher relative abundance of lactobacilli might not be of this particular strain, but it may influence the growth of others!

Round 2

Reviewer 1 Report

The work presented by Liu et al has been thoroughly revised and enriched with additional experiments concerning the tolerance of L. sakei  LZ217 to acid and temperature stressors. 
The weaknesses of the work in the first version have been removed, therefore, the manuscript can be accepted for publication. 

Minor remarks

Please give the names of the genus Lactobacillus in accordance with the new taxonomy.
The name Lactobacillus plantarum is still given in several parts of the manuscript. Please check. 

Please check Lacticaseibacillus rhamnoides (line 119).

Please check the figure number 2A (line 282)
